# Investigating the Sim2real Gap in Computer Vision for Robotics

## Abstract

A major challenge in designing machine learning systems for the real world is the sim2real gap, i.e. the change in performance when the system is transferred from simulation to the physical environment. Although many algorithms have been proposed to reduce this gap, it is not well understood. In this paper, we perform an empirical study of the sim2real gap for popular models in three standard computer vision tasks, monocular depth estimation, object detection, and image inpainting, in a robotic manipulation environment. We find that the lighting conditions significantly affect the gap for monocular depth estimation while object properties affect the gap for object detection and image inpainting, and these qualitative observations remain stable with different models and renderers.

## 1 Introduction

The goal of artificial intelligence is to create machines that can perform tasks in the real world. In early days of the field, behaviors were usually programmed into the machine. The field has since moved to a more data-driven paradigm, where the mapping from sensor data to action is learned. Doing so reduces the amount of domain knowledge that is needed but increases the number of necessary interactions with the real world, which creates additional challenges.

Collecting real world interactions is costly, both in terms of time and money, as robots must be reset at the end of each episode and may break if too many failures are encountered. A popular alternative approach is to create a simulation of the environment and robot and generate lots of interactions by running the simulation in parallel. However, when an agent learned in simulation is transferred to act in the real world, there is often a difference in performance because the simulation may not capture all aspects of the environment that are important for the task, even though practitioners take care to replicate the real world environment in the simulator. This difference is called the simulation-to-reality (sim2real) gap, and is often dealt with by training a robust and generalized agent on a large and diverse dataset (Zhao et al., 2020; Höfer et al., 2020) and validating the agent on diverse scenarios that are more realistic than those seen during training (Grau et al., 2022; Hagn & Grau, 2022).

To help guide this approach, we present a controlled experimental study to investigate the components of the sim2real gap, i.e. what environmental characteristics have the greatest effect on it. In doing so, we contribute a rigorous definition of the sim2real gap (Section 3). The conclusions would be relevant for designing both training and validation scenarios in simulation and in the real world. For example, if the number of cars on the road is shown to affect the gap for an autonomous driving system, the scenarios should include scenes on lonely country roads as well as busy city streets.

Intelligent agents consist of two modules: 1) perception, which gathers information about the environment and processes / analyzes it, and 2) control, which uses those results to select and carry out an action in the environment (Russell & Norvig, 2002). In this paper, we focus on the perception part, where the sensor is a RGB image; due to the ubiquity of cameras, many robotics pipelines are using perception from images. Specifically, we study the sim2real gap for three computer vision tasks applicable to robotics: 1) monocular depth estimation, 2) object detection, and 3) image inpainting.

We consider a robotic manipulation environment as the confined space makes it easier to control the various environmental characteristics. The environment consists of a robot manipulator on a pedestal next to a table with objects on it. We collect a dataset of RGBD images, randomizing various environmental characteristics of interest (such as lighting conditions and object properties). After creating a simulated version of the environment that closely approximates the physical environment, we generate RGBD images from the simulation that correspond to the real world images (Section 4). The paired RGBD images are then used to estimate the sim2real gaps for popular, SoTA models for the three tasks and statistical analysis is done to compute which environmental factors affect the gaps (Sections 3 and 5).

Our conclusions are task-dependent. The simulation underestimates the performance of the MiDaS DPT-Hybrid (monocular depth estimation) and EVA-LVIS (object detection) models but overestimates that of the LaMa (image inpainting) model. The light temperature and position affects the sim2real gap in monocular depth estimation, object geometry and texture affect the sim2real gap in object detection, and object texture affect the sim2real gap in image inpainting. The conclusions are also robust; for monocular depth estimation, if a different model or renderer is used, it remains that light properties affect the gap more strongly than object properties. Therefore, robotic systems whose perception modules contain a monocular depth estimation model should be validated with a wide variety of lighting conditions, while those containing an object detection model or image inpainting model should be validated with a wide variety of objects.

## 2 Related work

**Addressing the gap**    There have been many algorithms proposed to decrease the sim2real gap (Zhao et al., 2020; Höfer et al., 2020), with popular types of approaches being photo-realism, physical realism, domain randomization, and domain adaptation. Photo-realism and physical realism attempts to directly reduce the gap by making the geometry, materials, lighting, and physics in the simulator more closely resemble the real world, often utilizing hand-design and leveraging existing game engines (Martinez-Gonzalez et al., 2020; Roberts et al., 2022). Hagn & Grau (2022) takes it one step further by introducing errors into the simulator that decreases the distance between simulated and real world images.

Domain randomization and domain adaptation, aimed towards the agent rather than the simulator, generate data from varying simulator parameters (domains). Domain randomization aims to learn an agent that is robust to the gap by maximizing performance on a wide range of domains (Tobin et al., 2017; Muratore et al., 2022). On the other hand, domain adaptation enables the agent to ingest data from the domain at hand and adapt on-the-fly by utilizing domain-invariant features (Bousmalis et al., 2018; James et al., 2019) or features that can identify the domain (Beck et al., 2023). Another line of work proposes to utilize an estimate of the sim2real gap as a secondary learning signal, where the prediction is made by limited queries to the real world environment (Koos et al., 2013) or unseen domains (Muratore et al., 2021).

A common characteristic of a majority of methods is that they require identifying axes of variation in the environment and task that affect the sim2real gap. This is often done by intuition, but knowing what variations are most important would be useful for future work on the topic, e.g. what aspects of the simulator should be made more realistic.

**Understanding the gap**    Currently, our understanding of the sim2real gap appears to be primarily based on experience from transferring an agent trained in simulation to reality. According to Höfer et al. (2020), simulator physics for contact-rich tasks do not yet have high enough fidelity for domain randomization to work well, and effective sim2real algorithms for depth estimation require high-quality CAD models. Alghonaim & Johns (2021) identifies textures and background to be most important to randomize during domain randomization for pose estimation, but does not directly study the gap. In a similar vein, Sudhakar et al. (2023) studies the effect of introducing lighting variation and noise in object geometry and texture during training for object detection and instance segmentation and finds that noise in object geometry is the most important.

Some recent work has quantified the sim2real gap by measuring the change in performance when transferring an agent trained in simulation to reality, taking care that the simulation test environment approximates reality as much as possible (Stocco et al., 2022; Dieter et al., 2023). For example, Anderson et al. (2021)

and Kadian et al. (2020) create a 3D scan of the real world environment in simulation using a Matterport camera. Taking this one step further, Jaunet et al. (2021) provides a tool for visualizing the sim2real gap in an ego-localization task, highlighting areas of the map where the gap is largest and plotting the model features at those areas.

The works most similar in spirit to ours are Collins et al. (2019), which studies the fidelity of various simulators for modeling different types of robot arm motion, Rosser et al. (2020), which hypothesizes that certain morphology details affect the sim2real gap more than others for mechanical wing design, and Stocco et al. (2022), which studies the effect of image corruptions and adversarial examples on the sim2real gap in autonomous driving. However, no controlled and fine-grained studies of the effect of environmental properties are done in these papers.

**The related generalization gap** In concurrent work, Xie et al. (2023) conducts a similar study on a different but related quantity, the generalization gap, which is the change in the performance of an agent when environment properties are shifted from training to test, in either simulation or reality. For imitation learning on a robotic manipulation task, the authors observe that new camera positions and table textures lead to the largest generalization gap.

**Standardization of AI testing & automated driving** Automated driving addresses validation with a collection of different methodologies comprised of simulation, virtual testing, and real-world testing as well as clear scenario catalogues. The scenarios to be chosen should be relevant to the automated driving system and the relevant operational design domain and adequately cover a range of situations that are challenging for the system; a detailed approach for regulations is, for example, defined in Inland Transport Committee (2022). In particular, real-world testing is recommended for detecting issues that are not captured in simulation, i.e. the sim2real gap.

## 3 Approach

In order to analyze the sim2real gap, we begin by constructing a rigorous definition. As stated previously, the sim2real gap is the difference in the performances of a model when evaluated in the real world environment and in the simulation environment.

The performance of a model varies depending on characteristics of the environment, such as the lighting conditions, camera location, and object position. Therefore, when quantifying the sim2real gap, previous works (Anderson et al., 2021; Kadian et al., 2020) hold many of these characteristics constant between the real world and simulation, and we split those characteristics into two categories:

- $\mathbb{A}$: (Approximately) Equalizable in the two environments. Some characteristics can be measured with high accuracy in the real world and replicated in the simulator, such as the number of objects on the table and camera location. Other characteristics, such as object material and light temperature, can only be approximated by adjusting simulation parameters due to the structure of the simulator's physics or lighting model.

- $\mathbb{B}$: Not equalizable in the two environments. These characteristics may not be measurable in the real world with a high degree of accuracy or not controllable in the simulator, such as camera noise and inaccuracies in the simulator's lighting model, or even be unknown.

The sim2real gap is then a function of the characteristics in category $\mathbb{A}$, marginalizing over those characteristics in category $\mathbb{B}$. Mathematically, let the ground truth expected losses of a model in the real world environment when characteristics $\mathbf{a} \in \mathbb{A}$ are set equal to $\boldsymbol{a}$ be $\mathcal{L}_{real}(\mathbf{a} = \boldsymbol{a})$ and in the simulation environment be $\mathcal{L}_{sim}(\mathbf{a} = \boldsymbol{a})$. Then, the sim2real gap is defined as the function:

$$g(\boldsymbol{a}) = \mathcal{L}_{real}(\mathbf{a} = \boldsymbol{a}) - \mathcal{L}_{sim}(\mathbf{a} = \boldsymbol{a}) \tag{1}$$

In practice those losses are estimated using Monte Carlo after fixing the values of $\mathbf{a}$, and so we implicitly assume that the randomization is enough to cover the distribution of the characteristics in category $\mathbb{B}$ [1].

In our experiments, the characteristics in category $\mathbb{A}$ are the number of objects on the table, the geometry, texture, and position of the objects and robot, the camera configuration, and the position, temperature, and brightness of the light. Since the robot and camera set-ups are usually fixed before an agent is deployed, we vary only the other characteristics during data collection. For diverse values of $\boldsymbol{a}$, we collect RGBD images in the real world and in simulation while holding the characteristics in category $\mathbb{A}$ equal to $\boldsymbol{a}$ in the two environments. The data collection protocol is described further in the next section.

Given a task and model trained for that task, the real world images are used to estimate $\mathcal{L}_{real}(\mathbf{a} = \boldsymbol{a})$ and the simulation images $\mathcal{L}_{sim}(\mathbf{a} = \boldsymbol{a})$, resulting in estimates of the sim2real gap $\hat{g}(\boldsymbol{a})$. Because some characteristics in category $\mathbb{A}$ such as the texture of an object cannot be readily converted into numbers, we convert $\boldsymbol{a}$ into hand-designed features $\boldsymbol{f}$ (such as whether there is a metallic object on the table) and instead analyze the behavior of $\hat{g}(\boldsymbol{f})$ as a function of $\boldsymbol{f}$. By choosing a concise set of features that reflect the range of category $\mathbb{A}$ characteristics that are varied during data collection, analyzing $\hat{g}(\boldsymbol{f})$ leads to similar qualitative results as analyzing $\hat{g}(\boldsymbol{a})$. $\hat{g}(\boldsymbol{f})$ is modeled using a statistical model with uncertainty, thereby accounting for the stochasticity of the estimates and effect of characteristics in category $\mathbb{B}$ that are uncorrelated with those in category $\mathbb{A}$. If a certain feature is found to statistically significantly affect the sim2real gap $\hat{g}$, we may conclude that both validation scenarios and algorithms to reduce the gap should take care to cover the full range of that feature. More details are given in Section 5.1.

## 4 Dataset

In this section, we describe the real world and simulation environments (including how we equalize the category $\mathbb{A}$ characteristics described in the previous section) and the procedure used to collect the data.

### 4.1 Environment

We create a robotic hand manipulation arena, consisting of a Franka Emika Panda robot (Franka Emika) on a pedestal next to a table. Selected objects from the YCB dataset (Calli et al., 2017) are placed on the table at various positions and orientations, possibly stacked on top of one another. This object set was selected because it is popular in the robotics community, physical objects can be purchased, and textured meshes of objects are available online. The robot is stationary and does not interact with any other object. We next describe the physical and simulation environments; additional details can be found in Appendix A.2.

**Physical space**   Figure 1a shows a complete picture of the physical environment. The table is rectangular with blue metal legs, and the robot (which has a soft gripper (Soft Robotics)) is placed along one side on a blue metal mount with its black control box adjacent to the mount. There is a power surge opposite the robot on the table, blue metal posts at the corners on either side of the power surge, and a white tablecloth on top of the table. The RGBD camera, Azure Kinect (kin), is placed on a tripod near one of the corners of the table so that it has a clear view of the robot's gripper and tabletop; this sort of third-person view is common in the literature (Jangir et al., 2022). The camera position (relative to the center of the robot base) and orientation were computed using hand-eye calibration (Horaud & Dornaika, 1995) in ROS. The room is rectangular with white plaster walls, brown carpeting, and ambient fluorescent lighting, and we also have a LED floor lamp with controllable brightness and temperature settings to further illuminate the scene from one of two positions, either at the robot's left or the robot's right.

**Simulation space**   We approximate the physical space in the MuJoCo simulator (Todorov et al., 2012) using the robosuite (Zhu et al., 2020) package; an image is shown in Figure 1b with the default OpenGL renderer. Our simulation environment builds on robosuite's single-arm manipulation environment with a simulated Panda robot in a table arena. The table, robot, mount, control box, power surge, and posts were

---

[1]This assumption may not be true for camera artifacts if only one camera is available. Therefore, we view our results as being conditional on the camera we use, as well as on the general robot manipulation set-up.

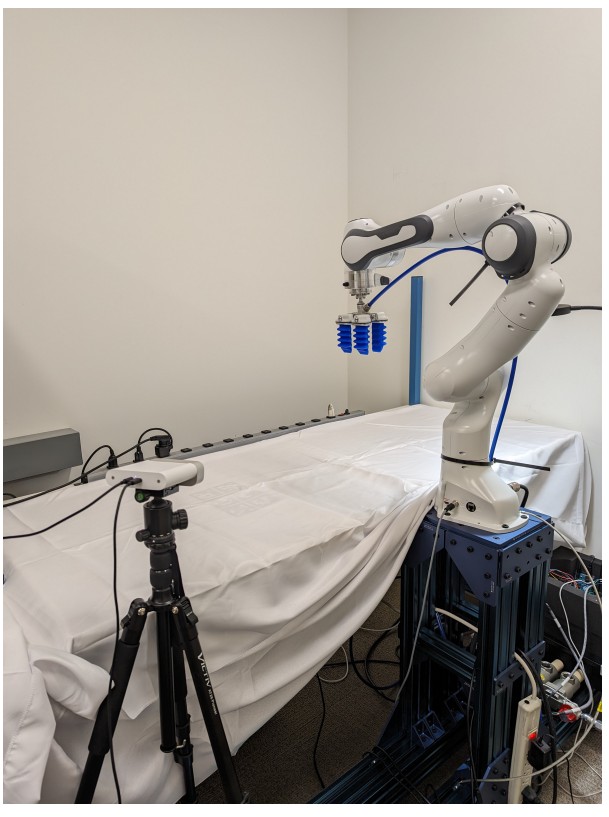
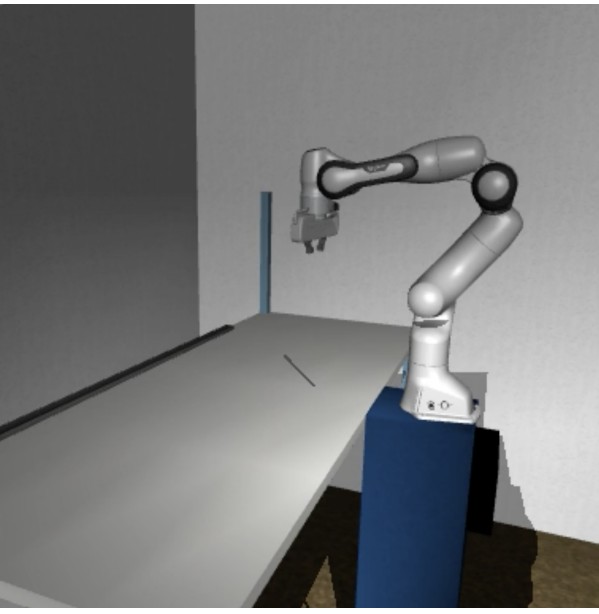

(a) Physical

(b) Simulation

Figure 1: Pictures of environments

scaled appropriately utilizing the sizes of the robot base in the physical space and in robosuite; their textures and material properties were chosen from those available in robosuite to mimic the physical environment. The camera FOV is set as 59 degrees, the same as the Azure Kinect, and its position was calculated by scaling the camera position in the physical space; its orientation is copied over without any change. For every possible brightness, temperature, and position setting of the lamp in the physical space, we find a corresponding setting for a light in the simulator. Its position and direction is approximated from the physical space and ambient, diffuse, and specular components are tuned so that an image of an upside down bowl looks similar to that in the physical space from the camera POV. Likewise, each YCB object mesh was scaled using the sizes of the table in the physical space and simulation and the size of the object in the physical environment; other material properties were chosen using Silicon Graphics, Inc. & Kilgard (1994).

### 4.2 Collection

The analysis dataset consists of pairs of RGBD images taken in the physical and simulation environments. For different pairs the YCB objects on the table, their positions and orientations, and the light position, brightness, and temperature are randomized, whereas within a pair they are equal.

Specifically, we repeat the following process three times for $N = 1, \ldots, 10$, where $N$ is the total number of objects on the table. Subject to the rules outlines in Appendix A.1.

- While there are fewer than $N$ objects on the table in the physical environment:
    - Sample an object $O$ uniformly at random. If that object is no longer available for placement (all instances are already on the table), redo.
    - Compile a list $\mathbb{L}$ of objects already on the table that $O$ can be stacked on; the list may be empty.

- Let $R$ denote the object that $O$ is stacked on. If $\mathbb{L}$ is empty, set $R = None$. Otherwise, select an object $R$ from $\mathbb{L}$, each with probability 0.1, or $R = None$ with probability $1 - 0.1|\mathbb{L}|$.
    - If $R \neq None$, no more objects can be stacked on it.
    - Uniformly randomly sample an orientation $Q$ for $O$. If $R = None$, also uniformly randomly sample a position $P$ on the table for $O$. Place object $O$ on the table; if it is not physically possible, redo.
    - Record $O$, $R$, $Q$, and $P$.

- For the lamp being on the left and right side of the robot and various temperature and brightness (eighteen settings in total): using the Open3D (Zhou et al., 2018) Azure Kinect Recorder and MKV Reader,

    - Record 1 second RGBD video.
    - Extract a single RGB image and corresponding depth from the video.

- Reset the simulation environment using the list of $O$, $R$, $Q$, and $P$. $Q$ is converted to a quaternion and $P$ is scaled in the same manner as the size of the robot base.

- For each of the eighteen lamp settings:

    - Adjust the simulation light properties to correspond to the current setting.
    - Extract a single RGB image and corresponding depth by taking a single step in the environment.

This results in a dataset of size $3 \times 10 \times 18 = 540$, where each data point consists of two RGBD images and a list of objects, stackings, their orientations and positions, and lamp setting.

## 4.3 Feature selection

During data collection, the YCB objects on the table, their positions and orientations, and the light position, brightness, and temperature were randomized. Instead of directly analyzing their effect on the sim2real gap, we design more interpretable features that serve as proxies and analyze the effect of these features on the gap. By selecting features that still capture information about the objects' geometry and texture and the light setting, the qualitative results of the analysis should not change. We compute the following features (sorted by type) for each data point.

- Object geometry

    - *size*: number of sampled objects, $N$
    - *boxes*: percentage of sampled objects with a box shape
    - *cylinders*: percentage of sampled objects that are cylindrical
    - *spheres*: percentage of sampled objects that are spherical
    - *nonconvex*: percentage of sampled objects that are nonconvex
    - *holes*: average number of holes in the sampled objects

- Object texture

    - *red*: percentage of sampled objects that are red
    - *orange*: percentage of sampled objects that are orange
    - *yellow*: percentage of sampled objects that are yellow
    - *green*: percentage of sampled objects that are green
    - *blue*: percentage of sampled objects that are blue
    - *purple*: percentage of sampled objects that are purple
    - *black*: percentage of sampled objects that are black
    - *white*: percentage of sampled objects that are white

– *gray*: percentage of sampled objects that are gray (metallic)

- Camera and light settings
  – *occluded*: 1 if one sampled object occludes another from the camera POV, 0 otherwise
  – *lightleft*: 1 if lamp is to the left of the robot from the camera POV, 0 otherwise
  – *lighttemp*: temperature setting of the lamp on a scale of $1 - 5$, higher means whiter light
  – *lightbright*: brightness setting of the lamp on a scale of $1 - 5$, higher means brighter

Further details on computing these features are found in Appendix A.3.

## 5 Analysis

We investigate the sim2real gap of popular, state-of-the-art models for three tasks, monocular depth estimation, object detection, and image inpainting. We also probe the sensitivity of our results to the simulator's renderer and the model architecture for monocular depth estimation. Before discussing the results in each task, we briefly describe the statistical model used in our analysis.

### 5.1 Statistical model

Suppose that we collect data according to the protocol in the previous section. Given a model, for each data point $i = 1, \ldots, 30, j = 1, \ldots, 18$ we compute its features $\boldsymbol{f}_{ij}$ and the estimated gap $\hat{g}_{ij} = \ell_{ij}^{real} - \ell_{ij}^{sim}$, where $\ell_{ij}^{real}$ is the loss of the model applied to the RGBD image from the physical environment and $\ell_{ij}^{sim}$ the loss of the model applied to the RGBD image from the simulation environment. Note that $i$ corresponds to the different YCB object settings and $j$ corresponds to the different lamp settings.

To model the estimated gap as a function of a set of $K$ features, we utilize a linear mixed model (Lindstrom & Bates, 1988) where the fixed effects are the features and the Gaussian random effects correspond to the thirty object settings. Specifically,

$$\hat{g}_{ij} = \alpha + \boldsymbol{\beta}^T \boldsymbol{f}_{ij} + \delta_i + \epsilon_{ij}, \quad \text{where} \quad \delta_i \sim \mathcal{N}(0, \sigma_g^2) \text{ and } \epsilon_{ij} \sim \mathcal{N}(0, \sigma_e^2) \tag{2}$$

where $\boldsymbol{\beta}$ and $\boldsymbol{f}_{ij}$ are $K$-dimensional vectors. This model accounts for correlations between RGBD images collected with the same table setting $i$ and variations caused by quantities not explicitly included in the model. Linear mixed effects models are optimized by maximum likelihood (Lindstrom & Bates, 1988) and we use the statsmodels (Seabold & Perktold, 2010) package; sometimes the optimization fails to converge, in which case we disregard the results. The optimization results are in the form of estimates $\hat{\beta}_k$ and standard errors of those estimates $\text{sd}(\hat{\beta}_k)$ for feature $k$. Following common practice in the statistics literature, those features for which $\hat{\beta}_k$ is statistically significantly different from zero, i.e. with large z-score $z_k = |\hat{\beta}_k|/\text{sd}(\hat{\beta}_k)$ and small p-value, are identified.

For each computed feature, we consider both the case where it is the only feature in the linear mixed effects model and the case where it is one of several features. If it is statistically significantly different from zero, in the former case we may conclude it affects the sim2real gap on average, while in the latter we may conclude it affects the sim2real gap conditional on the other features having certain values.

### 5.2 Monocular depth estimation

The first task is depth estimation from a RGB image, used in robotics for visual servoing. We study the DPT-Hybrid model (Ranftl et al., 2021) from MiDaS (Ranftl et al., 2022), which has good trade-off between speed and performance. The loss is the root normalized mean squared error of the per-pixel predictions of the reciprocal of the depth, after the predictions have been aligned to have similar scale and shift to the ground truth (Ranftl et al., 2022). [2]

---

[2]To normalize, we divide by the square of the mean of the reciprocal of the ground truth depth. Normalization deals with any scale differences between the simulation and the physical environment.

The average gap is negative; that is, evaluating the model on simulation images overestimates the loss on real world images. We first consider for each feature, the case where it is the lone fixed effect. *Lightleft* has a statistically significant (p-value < 0.1%) positive effect on the gap; a bar chart of the predicted gap as a function of the feature with a 95% confidence interval is shown in Figure 2a, overlaid with a scatterplot of the data. If the lamp is on the same side as the camera, the gap increases but decreases in magnitude, i.e. the over-estimation of the loss by the simulation environment becomes a slight under-estimation. *Lighttemp* has a statistically significant (p-value 0.9%) positive effect on the gap; the corresponding bar chart of the predicted gap as a function of the feature is shown in Figure 2b. If the lamp temperature increases, the gap decreases in magnitude, i.e. the over-estimation of the loss by the simulation environment improves. All optimization results are shown in Appendix B.

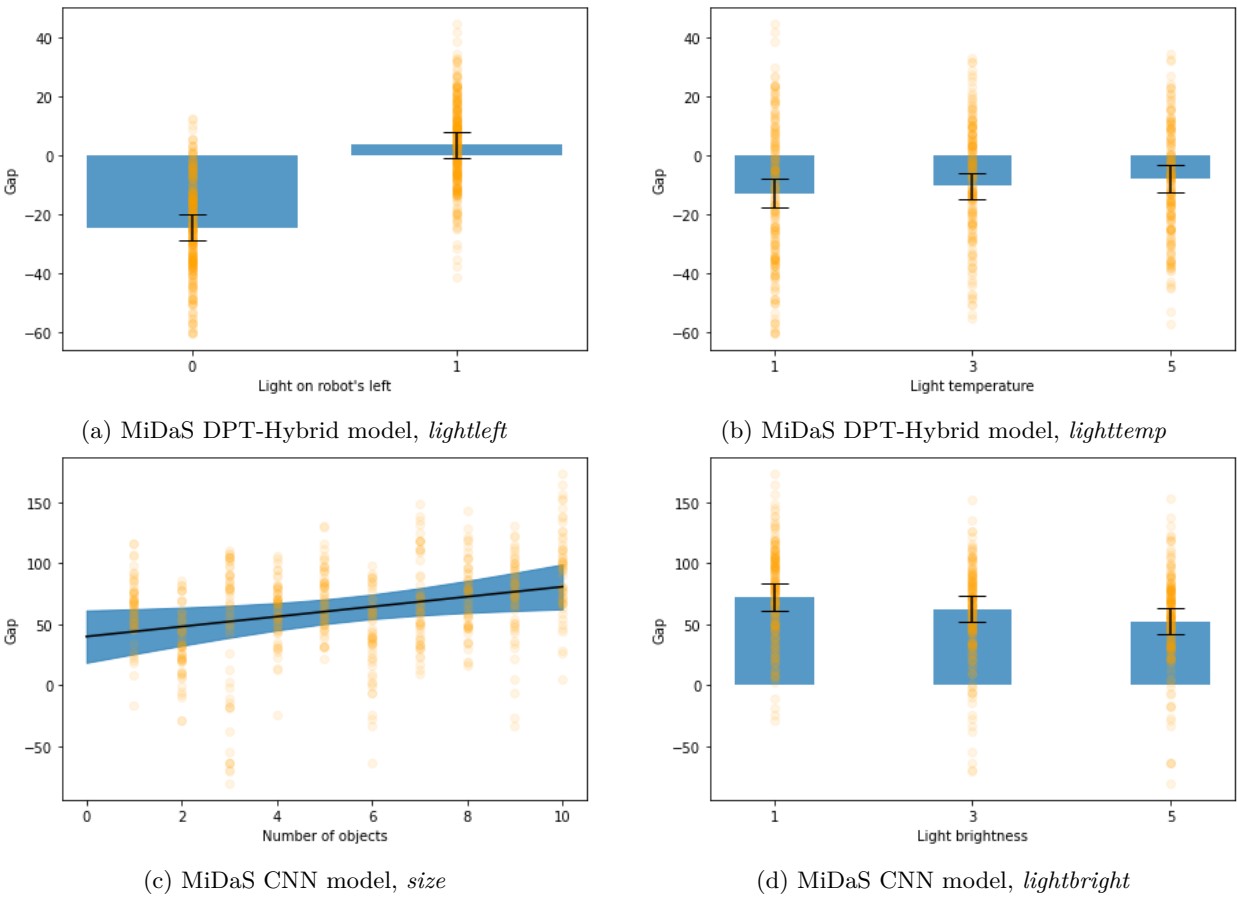

(a) MiDaS DPT-Hybrid model, *lightleft*

(b) MiDaS DPT-Hybrid model, *lighttemp*

(c) MiDaS CNN model, *size*

(d) MiDaS CNN model, *lightbright*

Figure 2: For the task of monocular depth estimation with the default OpenGL renderer, plots of the predicted gap (with 95% confidence interval and scatterplot of the data overlaid) as a function of a feature that is found to statistically significantly affect the gap when it is the only fixed effect in a linear mixed model. For better visualization, we use bar charts when the corresponding features have only a few possible values and the losses have been multiplied by 1000.

**Ablations**  To investigate how the conclusions may be affected by the experimental design choices, we also study 1) the original CNN model from MiDaS and 2) the DPT-Hybrid model under the NVISII renderer (Morrical et al., 2020), which utilizes ray-tracing.

For the CNN model, the average gap is instead positive but has greater magnitude than the average gap for the DPT-Hybrid model; this indicates that the DPT-Hybrid is more robust than the CNN model to domain shifts. When each factor is the lone fixed effect, *size* has a positive effect on the gap (p-value 2%), i.e. more objects on the table increases the magnitude of the gap; a plot of the predicted gap as a function of the

feature with a 95% confidence interval is shown in Figure 2c, overlaid with a scatterplot of the data. On the other hand, *lightbright* has a negative effect on the gap (p-value < 0.1%), i.e. having the lamp in the same area as the camera decreases the magnitude of the gap; the bar chart of the predicted gap as a function of the feature is shown in Figure 2d.

Next we consider groups of fixed effects/features by type. When there are only features describing the camera and light, *lightbright* has a negative effect on the gap (p-value < 0.1%). Figure 3 plots the predicted gap as a function of the light brightness with 95% confidence intervals, after fixing the other features at two different values seen in the dataset; changing those values do not seem to have a noticeable qualitative effect on the trend, only on the prediction magnitudes.

For the DPT-Hybrid model with the NVISII renderer, *lightleft* and *lighttemp* affect the gap, with very similar effect estimates and p-values to those for the same model with the default OpenGL renderer. However, *yellow* now has a statistically significant positive effect on the gap (p-value 3.1%); a plot of the predicted gap as a function of the feature with a 95% confidence interval is shown in Figure 4a, overlaid with a scatterplot of the data. Similar plots for *lightleft* and *lighttemp* are shown in Figures 4b and 4c. Overall, the light properties are more relevant for the sim2real gap in all cases, as the p-values for those features are smaller than those of the other statistically significant features.

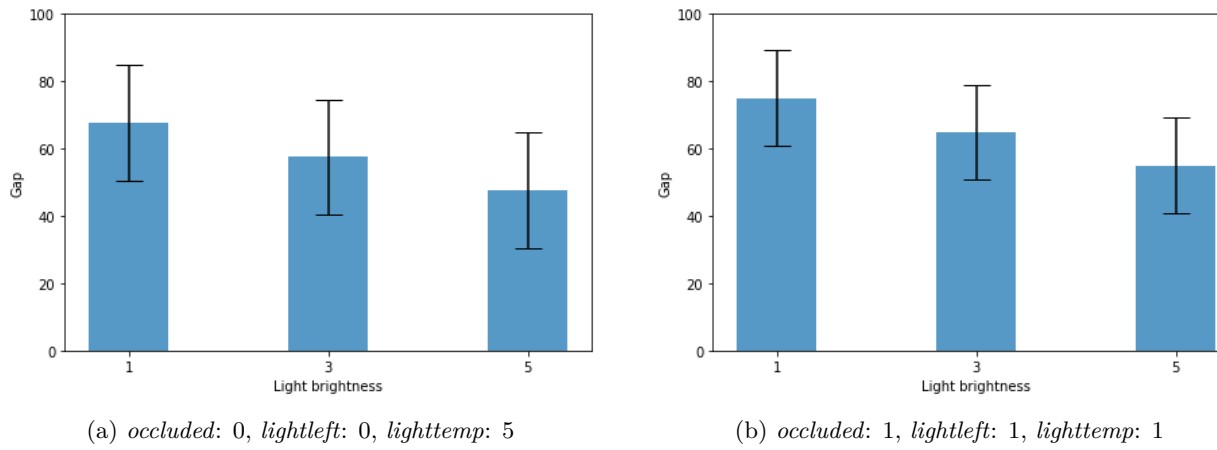

(a) *occluded*: 0, *lightleft*: 0, *lighttemp*: 5            (b) *occluded*: 1, *lightleft*: 1, *lighttemp*: 1

Figure 3: For the MiDaS CNN model with the default OpenGL renderer, bar charts of the predicted gap (with 95% confidence interval) as a function of the statistically significant feature *lightbright* when only camera and light features are included. The other features have been fixed at certain values seen in the dataset. As above, the losses have been multiplied by 1000.

## 5.3 Object detection

The next task is object detection, which is utilized in robotics for visual understanding of the robot's surroundings. We study the state-of-the-art EVA model (Fang et al., 2022) that is fine-tuned on the LVIS dataset, because its categories include most of the YCB objects we selected. This model produces object category and bounding box proposals. Since we cannot obtain ground truth bounding boxes in the exact manner of the LVIS dataset, we define the loss as the percentage of the $N$ sampled objects for which no proposal has the same category.

As in the previous subsection, the average gap is negative, and we first consider the cases where there is a single fixed effect/feature. *Holes* appears to have a positive effect on the gap (p-value 0.7%); that is, more holes in the objects reduces the overestimation of the loss. The same is true of more gray/metallic objects (p-value 1.6%), but more red objects on the table has a negative effect on the gap (p-value < 0.1%). Figures 5a – 5c plot the predicted gap as a function of each of these features with 95% confidence intervals and a scatterplot of the data overlaid.

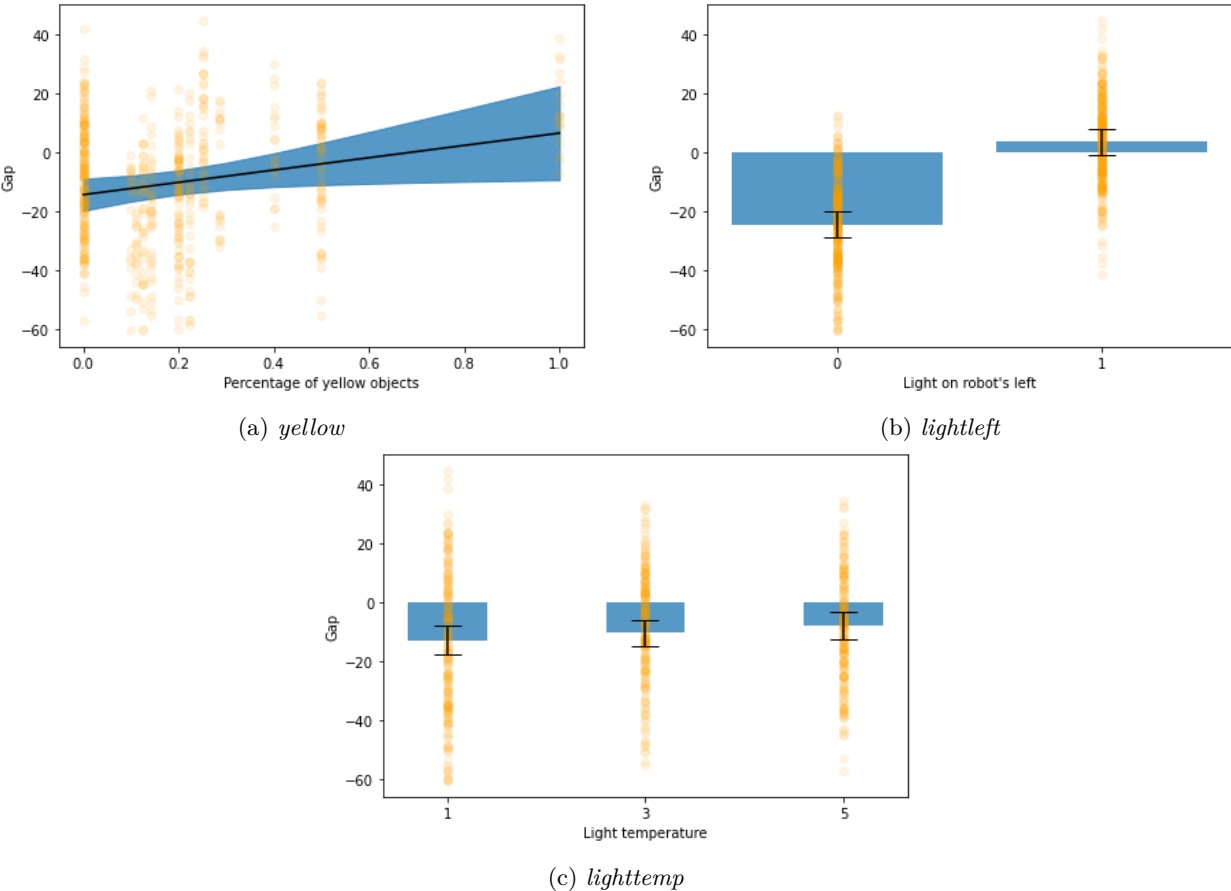

(a) *yellow*  (b) *lightleft*

(c) *lighttemp*

Figure 4: For the MiDaS DPT-Hybrid model with the NVISII renderer, plots of the predicted gap (with 95% confidence interval and scatterplot of the data overlaid) as a function of a feature that is found to statistically significantly affect the gap when it is the only fixed effect in a linear mixed model. As above, the losses have been multiplied by 1000.

Next we consider groups of fixed effects/features by type. When there are only features describing the object geometries, *boxes*, *spheres*, *nonconvex*, and *holes* all have statistically significant (p-value < 0.1%) positive effects on the gap. Figure 6 plots the predicted gap as a function of *holes* with 95% confidence intervals, after fixing the other features at values seen in the dataset [3]. There does not seem to be a qualitative difference in the trends. However, it appears that the uncertainty in the predicted gap is lower in the second subfigure especially when *holes* is small, as in most of the dataset; this is not surprising since the number of objects on the table is larger. When there are only factors describing the object textures [4], *red* has a statistically significant (p-value 0.8%) negative effect on the gap. When there are only factors describing the camera and light, there are no conclusions that can be drawn. Overall, we may conclude that the object geometries and colors affect the sim2real gap in object detection and not the camera or light properties. All optimization results are shown in Appendix B.

## 5.4 Image inpainting

The final task is image inpainting, which is helpful in robotics when there is occlusion by nuisance objects; we study the popular LaMa model (Suvorov et al., 2021). The loss is the expected squared error of a predicted

---

[3]We do not create the same figures for the other significant features as their values create restrictions on those of other features.

[4]We exclude *white* to avoid multicollinearity.

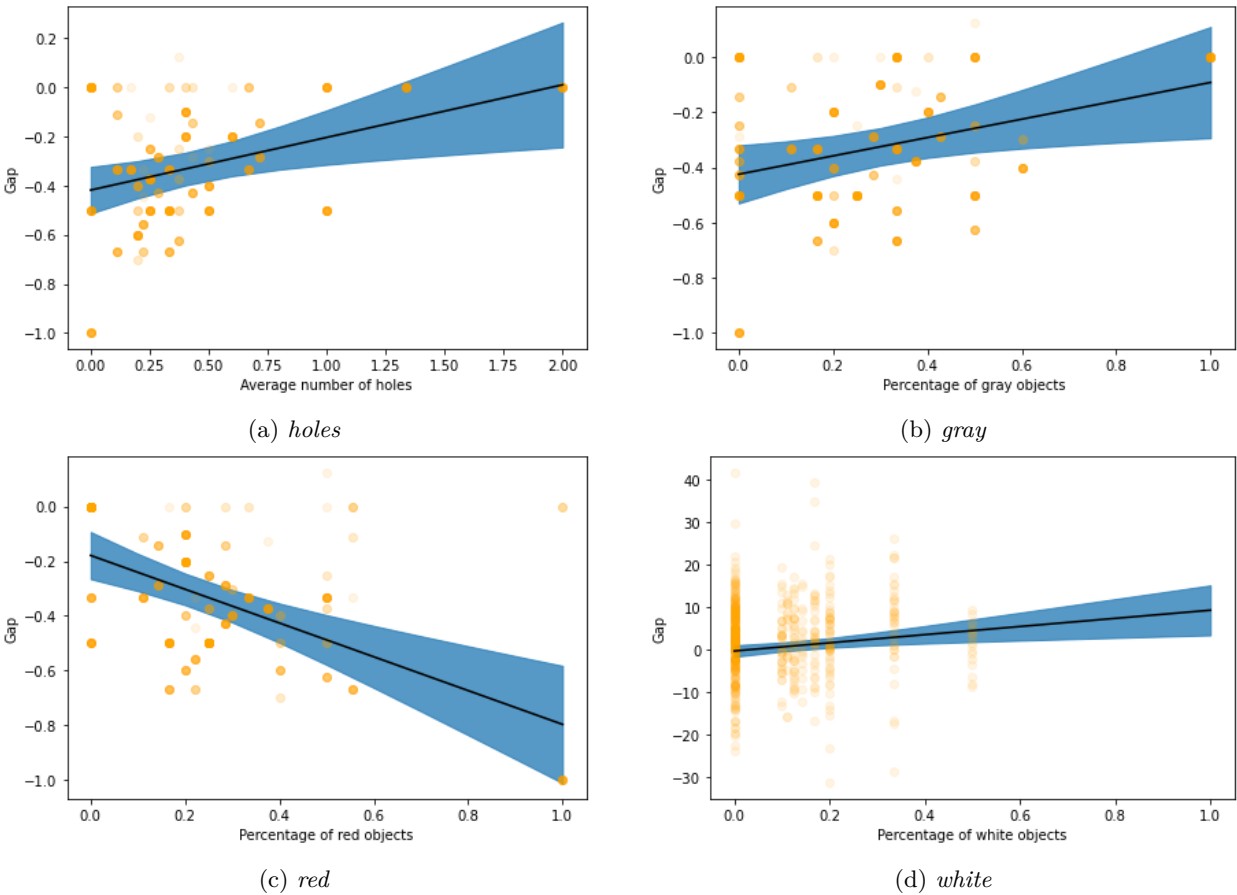

Figure 5: Line plots of the predicted gap (with 95% confidence interval and scatterplot of the data overlaid) as a function of a feature that is found to statistically significantly affect the gap when it is the only fixed effect in a linear mixed model. (a)-(c) are for object detection and (d) is for image inpainting, with the losses multiplied by 1000.

pixel when some in the image are randomly masked. In particular, for each data point, $\mathcal{L}_{real}$ is estimated as the MSE of the pixel predictions, averaged over 10 random masks on the RGB image from the physical environment; $\mathcal{L}_{sim}$ is estimated similarly. The masks are sets of rectangles and polygonal chains with large width, making simple interpolation ineffective.

The average gap is positive, implying that the model performs worse on real world images than on simulation images. When there is a single fixed effect/factor, *white* has a statistically significant (p-value 0.5%) positive effect. Figure 5d plots the predicted gap as a function of *white* with a 95% confidence interval and a scatterplot of the data overlaid. As with object detection, only the object textures affect the sim2real gap. The optimization results are shown in Appendix B.

### 5.5 Discussion

To sum up, for a SOTA monocular depth estimation model the simulation environment underestimates the performance, which is ameliorated by placing the lamp near the camera and increasing the temperature. When a different renderer is used, increasing the number of yellow objects also ameliorates the gap. When an older model is used, the simulation environment overestimates the performance, which is exacerbated by increasing the number of objects and ameliorated by increasing the brightness of the lamp. For object detection the simulation environment also underestimates the model performance, which is instead exacer-

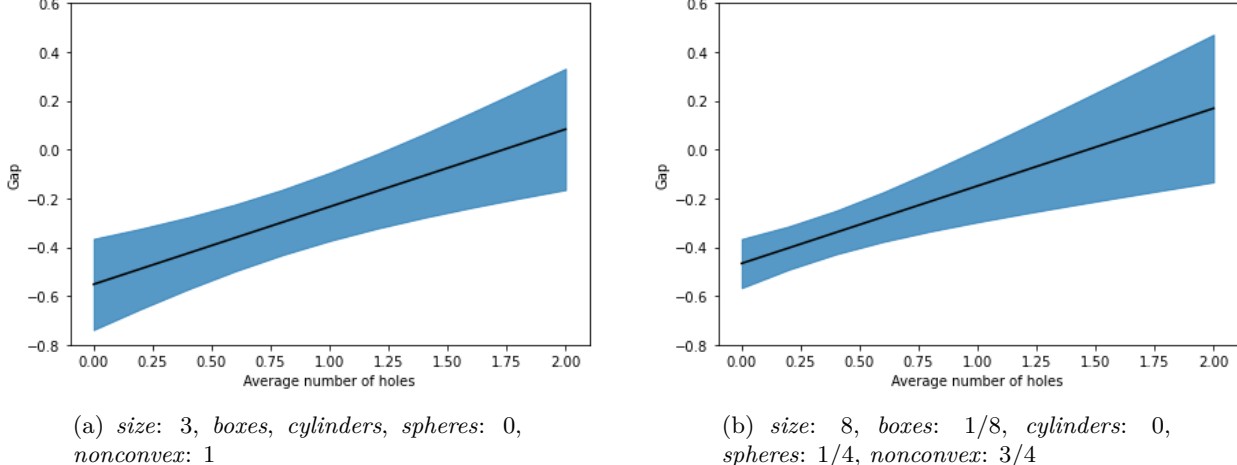

(a) *size*: 3, *boxes*, *cylinders*, *spheres*: 0, *nonconvex*: 1

(b) *size*: 8, *boxes*: 1/8, *cylinders*: 0, *spheres*: 1/4, *nonconvex*: 3/4

Figure 6: For object detection with the default OpenGL renderer, line plots of the predicted gap (with 95% confidence interval) as a function of the statistically significant feature *holes* when only object geometry and layout features are included. The other features have been fixed at certain values seen in the dataset.

bated by more red objects and diminished by more holes in the objects. For image inpainting the simulation environment overestimates the model performance, which is exacerbated by more white objects.

In general, light properties have a greater impact on depth estimation but object properties have a greater impact on object detection and image inpainting. Therefore, when validating depth estimation models it is important to consider diverse lighting conditions and have an accurate lighting model, while when validating object detection models it is important to consider diverse and accurate object geometries and textures. The behavior of the sim2real gap appears to be highly task-dependent, explaining common practices such as hand-selecting variables for domain randomization.

## 6 Conclusion

In this work, we conducted a systematic investigation of factors that impact the sim2real gap in computer vision for robotic manipulation, focusing on popular models for monocular depth estimation, object detection, and image inpainting. After creating a replica of the physical environment in robosuite (Zhu et al., 2020) that consists of objects from the YCB dataset (Calli et al., 2017) on a table and a Panda manipulator (Franka Emika), we collected a dataset of RGBD images with diverse objects and lighting conditions. Statistical analyses show that light properties affect the gap more than object geometry and texture in monocular depth estimation while the opposite is true in object detection and image inpainting. A natural extension would be to study both perception and control by considering simple manipulation tasks such as pick-and-place and introducing physics-related factors such as object weight and material friction.

**Broader Impact Statement**

The conclusions in this paper can inform the creation of any robotic system that incorporates computer vision. As such, our work does not have direct societal impact, but those systems may either have positive impact or negative impact.

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
