# OpenReview forum: "Investigating the Sim2real Gap in Computer Vision for Robotics"
_TMLR — Rejected by TMLR_

### Review · Reviewer_eMJ2 · 2024-02-19

**Summary Of Contributions:**

The authors measure the performance gap between simulation and reality of several CV models on three different tasks: monocular depth estimation, object detection and image inpainting.

To do so, they collect data both in the real world, and from rendering a scene that re-creates the real world setting.

They then evaluate the models while varying certain factors in both reality and simulation.

**Audience:**

No

**Claims And Evidence:**

No

**Requested Changes:**

As described above, I believe the experimental setup is fundamentally flawed. A possible fix would be to measure the performance difference between the test-data split of the training data, and the real world. However, this also requires very careful attention to detail as one has to justify why one expects the model to generalize to that particular real situation.

**Strengths And Weaknesses:**

If I understand the paper correctly, the authors do not train the evaluated models on their collected data, but take pre-trained models and evaluate them on their collected data from both the real world and simulation.
However, this is a fundamental misunderstanding of what the sim2real gap is meant to express because in their setup, *both* the simulated data and the real data are different from the training data, so the gap between both is meaningless. What we're really interested in is the gap between the *training* environment (which typically is in simulation) and deployment environment.
This also explains their result that in some settings they find that the model actually performs better in the real world than in simulation (i.e. having a 'negative' sim2real gap).

Hence, I cannot recommend acceptance.

On another point: I find the paper overall too long and including too many (irrelevant) details that are specific to their very specific experimental setup. In a future, revised, version of their work, I'd recommend trying to distill more general take-aways from their experiments and present those more concisely.

---

> ### Author Response · Authors · 2024-04-12
>
> Thank you to the reviewer for your comments.
>
> We think that there is some confusion with the definition of the sim2real gap, which we will clarify in future versions of the paper.
> There are three environments in a machine learning pipeline:
> - simulation environment used for training a model,
> - simulation environment used for testing a model that mimics the real world environment, and
> - real world environment.
> The sim2real gap is defined as the difference in the model performance between Environment 2 and Environment 3 (Rosser et al.), and this is what we study.
>
> In many cases, Environment 1 and Environment 2 are the same, so the sim2real gap is defined as the difference in model performance between Environment 1 and Environment 3, as you state.
> But that may not always be the case.
> Sometimes practitioners may not be able to train a large model themselves (due to monetary, time, or expertise constraints) and thus apply a pre-trained model after testing scenarios in their environment-specific simulator.
> However, to ensure that there are not scenarios where the model fails (i.e. where the sim2real gap is too large), testing must also be done in real world scenarios.
> Our work provides guidance on how to select these real world scenarios; for example, when applying the MiDaS DPT-Hybrid model, we advise to consider a wide range of lighting conditions.
>
> We agree that the case where Environment 1 and Environment 2 are the same is also important to include.
> Therefore, we have conducted an additional analysis on a MiDas CNN model trained in simulation, which we will include in future versions of the paper.
> The details and results are in a separate comment.
> In future versions of the paper we will also include analyses on object detection and image inpainting models trained in simulation.
>
> [Rosser et al.]: Sim2real gap is non-monotonic with robot complexity for morphology-in-the-loop flapping wing design, ICRA 2020.

---

### Review · Reviewer_CKUS · 2024-03-15

**Summary Of Contributions:**

The authors record a dataset of matching sim and real images, then fit a linear model to this data and use that to make statements about sytematic issues of visual sim2real transfer.

**Audience:**

No

**Broader Impact Concerns:**

No concerns.

**Claims And Evidence:**

No

**Requested Changes:**

1. The authors need to either remove the claims to generality or conduct a lot more experiments across scenes, objects, camera models, renderers, etc.
2. I'd love if the authors would try different statistical models and not just a linear mixed model.
3. I'd recommend the authors include a section with clear recommendations based on their work - what claims can they make based on this work that are generally true for the sim2real gap?
4. I'd remove the robot from the simulation for faster rendering.
5. I'd recommend adding a whole lot more detail about how the authors match sim and real to begin with.
6. I'd do a pass on the writing of the paper and check for weird expressions and easy readability.
7. I'd explain the choices that went into the scope of this paper a bit more.

**Strengths And Weaknesses:**

### Strengths

- **S.1**: The authors created a matching sim and real dataset, which takes a lot of time

### Weaknesses

- **W.1**: The authors lay claim to a "systematic investigation of factors that impact the sim2real gap in computer vision for robotic manipulation" and I absolutely don't see that here. For me, the sim2real gap comes not just from different factors in a given environment but also from the very specific hardware and software used. I don't understand how the authors can claim any sort of generality without having tried out for example different cameras (to determine the same for the camera) or different scenes (e.g. different sets of objects, not just YCB).
- **W.2**: I don't understand how the authors can fit a linear model to their data when in many cases (esp. Fig.5 a, b, c, d) the data looks anything but linear.
- **W.3**: Even in their own environment, I don't understand how the conclusions drawn are useful to other researchers. The discussion says for example "When a different renderer is used, increasing the number of yellow objects also ameliorates the gap" - how is this not hyper-specific to your one renderer, camera position, task, model, and object set? How can I apply this knowledge to my own sim2real experiments?
- **W.4**: I don't understand why there's a robot in the scene at all. Is the robot used for anything?
- **W.5**: Many crucial details that could make this more reproducible and useful are glossed over. For example page 5 mentions matching the real lighting and the simulator lighting "...so that an image of an upside down bowl looks similar..." - what does that mean? I feel like this tuning is very crucial to how your model is affected by lighting. How do you extract a single image from the 10 video frames? Do you take a random key frame? Do you take the mean? Do you take the median? Do you take any random frame? Do you take the first, the last? These are all little design decisions that can impact the effects of sim2real transfer and they need to be very clearly specified.
- **W.6**: The writing is a bit weird in places. For example I think on page 4, you mean a "power bar" or a "power strip", not a "power surge" which is a very different thing.
- **W.7**: Why were these specific models chosen (e.g. why LaMa, why EVA, etc.)? And why is there image in-painting? Have you seen this being used in a real robotic setting before or is this a guess?

---

> ### Author Response · Authors · 2024-04-12
>
> Thank you to the reviewer for your comments.
> We will take care in future versions of the paper to explain details of the design choices and be more circumspect about the claims.
>
> ### Main takeaways ###
> We agree that the particular features found to statistically significantly affect the gap are probably specific to the setup.
> However, we believe that we can take away certain general observations.
> First, the gap may not always be positive - it may be that the particular real world scenario considered exactly suits the model - so it is important for practitioners to conduct a case-by-case analysis.
> Second, camera and light properties affect the gap more strongly than object properties for monocular depth estimation, and therefore practitioners should include a wide range of camera and lighting conditions in their real world tests.
> We found this to hold while changing the architecture of the model, the renderer, and the data that the model is trained on (please see the separate comment).
> We plan to replicate these changes for object detection and image inpainting in future versions of the paper.
>
> ### Paper scope and impact ###
> Although we consider a specific (but realistic setup), we achieved interesting results for robotics tasks. It is true that more scenarios etc. would be very beneficial, but we have addressed a subset to enable better control over the simulation and physical environments and thus to better understand the sim2real gap.
> Prior works have shown that camera properties do affect the sim2real gap and provided ways to mitigate it by transforming simulation data to mimic real world data (Hagn and Grau).
> In contrast, our work considers object and light properties and is able to provide guidance on how to collect that data, which may be expensive and time-consuming.
>
> ### Linear mixed effects ###
> The linear mixed effects model allows for good interpretation of the results.
> For example, when there is only one fixed effect, the linear mixed effects model tests whether there are differences in the gap for the various values of the fixed effect, after adjusting for correlated errors arising from observations from the same object setting.
> Moreover, because of the correlated errors, scatterplots may not clearly show patterns in the data.
>
> ### Robot in the scene ###
> Our final research goal is to extend this analysis to downstream manipulation tasks, which is why there is a robot in the scene, but we decided to first focus on the perceptual modules.
> The outputs of perceptual modules like depth estimation and object detection have been shown to be very useful for robotics (Hofer et al.) and are utilized in other applications like scene recognition.
>
> ### Model choices ###
> The models were chosen based on two criteria: 1) state-of-the-art, or close to it at the time of selection, which was early 2023, and 2) pre-trained models are available online.
> Image inpainting has indeed been considered in robotics as a method to deal with highly dynamic videos, for example (Bescos et al.).
>
> ### References ###
> - [Hagn and Grau] Optimized data synthesis for dnn training and validation by sensor artifact simulation, 2022.
> - [Hofer et al.] Perspectives on Sim2Real Transfer for Robotics: A Summary of the RSS 2020 Workshop.
> - [Bescos et al.] Empty Cities: Image Inpainting for a Dynamic-Object-Invariant Space, ICRA 2019.

---

### Review · Reviewer_wqYS · 2024-04-01

**Summary Of Contributions:**

This paper conducts an analysis on the performance difference of off-the-shelf computer vision models on images captured from simulators vs the real world. Experiments identify several variables of varying importance that contribute to the difference in performance.

**Audience:**

No

**Broader Impact Concerns:**

Similar to my last point in the weaknesses section. The experimental setup is very niche, and since the analysis is also very setup specific, none of the conclusions in this paper should be intepreted beyond the exact problem setup.

**Claims And Evidence:**

No

**Requested Changes:**

Most of my concerns are included in the above section. TL;DR:
1. Improve writing
2. Either significantly increase scope of environments, or signal in the title and intro that the paper only studies a very niche setup.
3. Train the model with data from the simulator and properly measure the sim2real gap.

**Strengths And Weaknesses:**

This paper is well motivated. However, there are too many major weaknesses which I believe make the paper subpar for acceptance.

- Writing quality.
  - There are grammatical errors throughout the paper. E.g. “There have been algorithms … -> There are algorithms …”
  - The related work section is very disorganized. The bullet points seem arbitrary and do not represent good partitions of the related work.
  - The term “automated driving” is confusing – what does it represent? A specific type of approach or “self-driving”?
- Methodology
  - The authors use off-the-shelf, pretrained models to evaluate “sim2real” gap from images captured in real-world and from the simulator.
This is not the sim2real gap the community is interested to study, but rather a domain gap on the superficial level. To properly evaluate the sim2real gap, the authors should 1. Train the model using sensor data collected from the simulator and then 2. Measure how the model performs in the real-world vs. in the simulator.
  - The authors aim to study the sim2real gap in computer vision for robotics, but only measure performance of perception modules. I prefer to see downstream metrics directly, since perception metrics are means to proxy and might not reflect the performance of the entire perception-control stack.
  - misc: The authors mention there is not a good way to label the collected images as LVIS, which is why they choose to only evaluate proposal. I wonder why is that? LVIS should have explanation on their labeling criteria.
- Title and introduction
  - The title and introduction suggest the paper studies sim2real gap on a generic level, but in reality the execution only covers a very specific setup.

---

> ### Author Response · Authors · 2024-04-12
>
> Thank you to the reviewer for your comments.
> We will incorporate your suggestions on writing into future versions of the paper.
>
> ### Sim2real gap definition ###
> We think that there is some confusion with the definition of the sim2real gap, which we will clarify in future versions of the paper.
> There are three environments in a machine learning pipeline:
> - simulation environment used for training a model,
> - simulation environment used for testing a model that mimics the real world environment, and
> - real world environment.
>
> The sim2real gap is defined as the difference in the model performance between Environment 2 and Environment 3 (Rosser et al.), and this is what we study.
>
> In many cases, Environment 1 and Environment 2 are the same, so the sim2real gap is defined as the difference in model performance between Environment 1 and Environment 3, as you state.
> But that may not always be the case.
> Sometimes practitioners may not be able to train a large model themselves (due to monetary, time, or expertise constraints) and thus apply a pre-trained model after testing scenarios in their environment-specific simulator.
> However, to ensure that there are not scenarios where the model fails (i.e. where the sim2real gap is too large), testing must also be done in real world scenarios.
> Our work provides guidance on how to select these real world scenarios; for example, when applying the MiDaS DPT-Hybrid model, we advise to consider a wide range of lighting conditions.
>
> We agree that the case where Environment 1 and Environment 2 are the same is also important to include.
> Therefore, we have conducted an additional analysis on a MiDas CNN model trained in simulation, which we will include in future versions of the paper.
> The details and results are in a separate comment.
> In future versions of the paper we will also include analyses on object detection and image inpainting models trained in simulation.
>
> ### Paper scope and impact ###
> Our final research goal is to extend this analysis to downstream control tasks, but we decided to first focus on the perceptual modules.
> The outputs of perceptual modules like depth estimation and object detection have been shown to be directly useful for robotics (Hofer et al.) and are utilized in other applications like scene recognition.
> Further, we believe that the focus on one stage of the pipeline is valuable as the knowledge gained can also improve the overall pipeline.
>
> Our work can be seen as limited or it can be seen as a start in order to highlight the problem area. Although we consider a specific (but realistic setup), we achieved interesting results for robotics tasks. It is true that more scenarios etc. would be very beneficial, but we have addressed a subset to enable better control over the simulation and physical environments and thus to better understand the sim2real gap.
> We provide results in this setup and hope to inspire others to expand on our research.
>
> ### Bounding boxes for object detection ###
>
> According to the LVIS paper (Gupta et al.), the bounding boxes were created through a multi-step process involving different types of labeling by crowd workers and verification steps by other crowd workers.
> Since there are no quantitative criteria for the labels aside from the crowd workers' judgements, we're not sure how well our annotations would align with theirs.
> However, in future versions of the paper, we will include an object localization metric as well based on our own annotations.
>
> ### References ###
> - [Rosser et al.]: Sim2real gap is non-monotonic with robot complexity for morphology-in-the-loop flapping wing design, ICRA 2020.
> - [Hofer et al.] Perspectives on Sim2Real Transfer for Robotics: A Summary of the RSS 2020 Workshop.
> - [Gupta et al.] LVIS: A Dataset for Large Vocabulary Instance Segmentation, arXiV 2019.

---

### Author Response · Authors · 2024-04-12
**Analysis on a model trained on simulation environment**

We conducted the same analysis of the sim2real gap for a MiDaS CNN model trained only on data from the simulation environment, consisting of 25,000 RGBD images.
To generate the data, we randomly place objects in the same way as the dataset described in the paper, and randomize the parameters of the cameras, lights, and the material properties of all objects in the scene (mimicking domain randomization).
To train the model, we follow the process outlined in MiDaS (Ranftl, Lasinger, et al.)

The average gap is positive, like the pre-trained MiDaS CNN model.
*Lightleft* has a statistically significant negative effect on the gap (p-value 2.2%) and *lighttemp* has a statistically significant positive effect on the gap (p-value <0.1%) when they are the lone fixed effect.
When all features describing the camera and light are included (and when all features are included), as before *lightleft* has a negative effect on the gap and *lighttemp* has a positive effect on the gap.
Although the specific features are different, the main observation stated at the end of Section 5.2 still applies; the camera and light properties strongly affect the gap for monocular depth estimation.

[Ranftl, Lasinger, et al.] Towards Robust Monocular Depth Estimation: Mixing Datasets for Zero-shot Cross-dataset Transfer, TPAMI 2022.

---

### Decision · Action_Editor_89Wy · 2024-05-27

**Recommendation:** Reject

**Comment:**

Reviewers are unanimous in their recommendations. In particular, the paper is not extensively evaluating the right type of sim-to-real gap that would be useful for robot manipulation. Significant updates are needed to improve the paper. Even for the task of depth estimation, as far as I can tell, no transparent objects were included, which are a challenge for RGBD sensor simulation, and an area where we expect the sim-to-real-gap to be high.  As such, unfortunately, the paper needs significant refocus and execution improvements before it is useful to the robotics+ML community. The motivation of the paper is good, but in order for the paper to be useful, it needs to leads to actionable information that is used in the robust training of policies, or in system identification. We encourage the authors to continue improving this line of work in the future.

**Audience:**

The paper is primarily of interest to robotics and ML researchers.

**Claims And Evidence:**

While the paper set out to characterize sim-to-real gap, it did so for very constrained settings for perception tasks such as monocular depth estimation and object detection, and did not evaluate the gap between a behavior learned in simulation in the real world, nor did the gap in dynamics was studied.